# The macroecological dynamics of sojourn trajectories in the human gut microbiome

William R. Shoemaker,[1] Jacopo Grilli[1]

**ABSTRACT**    The human gut microbiome is a dynamic ecosystem. Host behaviors (e.g., diet) provide a regular source of environmental variation that induces fluctuations in the abundances of resident microbiota. Despite these displacements, microbial community members remain highly resilient. Population abundances tend to fluctuate around a characteristic steady-state over long timescales in healthy human hosts. These temporary excursions from steady-state abundances, known as sojourn trajectories, have the potential to inform our understanding of the fundamental dynamics of the microbiome. However, to our knowledge, the macroecology of sojourn trajectories has yet to be systematically characterized. In this study, we leverage theoretical tools from the study of random walks to characterize the duration of sojourn trajectories, their shape, and the degree that diverse community members exhibit similar qualitative and quantitative dynamics. We apply the stochastic logistic model as a theoretical lens for interpreting our empirical observations. We find that the typical timescale of a sojourn trajectory does not depend on the mean abundance of a community member (i.e., carrying capacity), although it is strongly related to its coefficient of variation (i.e., environmental noise). This work provides fundamental insight into the dynamics, timescales, and fluctuations exhibited by diverse microbial communities.

**IMPORTANCE**  Microorganisms in the human gut often fluctuate around a characteristic abundance in healthy hosts over extended periods of time. These typical abundances can be viewed as steady states, meaning that fluctuating abundances do not continue towards extinction or dominance but rather return to a specific value over a typical timescale. Here, we empirically characterize the (i) length (i.e., number of days), (ii) relationship between length and height, and (iii) typical deviation of a sojourn trajectory. These three patterns can be explained and unified through an established minimal model of ecological dynamics, the stochastic logistic model of growth.

**KEYWORDS**   microbiome, macroecology, ecological dynamics, human gut microbiome, microbial dynamics, microbial communities, biological physics, ecology, microbial ecology

Microbial communities in the human gut often maintain steady-state abundances over extended timescales. This tendency has been the subject of observational macroecological investigations (1) as well as experimental efforts imposing external perturbations (2, 3). However, we lack a quantitative understanding of the dynamics of microorganisms fluctuating around their steady state. Excursions away from, and subsequent return to, said steady-state values, known as sojourn trajectories, have been systematically studied for various stochastic models (4, 5) and quantitatively examined in neurological (6) and geophysical systems (7). Such an approach may also aid our understanding of microbial community dynamics.

**Peer Reviewer** María Rebolleda Gómez, University of California, Irvine, Irvine, California, USA

Address correspondence to William R. Shoemaker, williamrshoemaker@gmail.com.

The authors declare no conflict of interest.

See the funding table on p. 5.

10.1128/msystems.01221-25 **1**

Here, we characterize the sojourn trajectories of microbial communities in the human gut, investigating the duration of sojourn trajectories ($\mathcal{T}$) and how this quantity relates to the typical deviation from steady-state abundance. We interpret our findings and generate quantitative predictions using the stochastic logistic model (SLM) of growth, a minimal ecological model where species dynamics are governed by self-limiting growth and environmental noise. This choice of model is reasonable as it has been successfully applied to investigate the macroecology of microbial communities within and across human hosts (8, 9), community-level measures of ecological distances and dissimilarities (10), the temporal dynamics of alternative stable states (1), experimental communities (11), dynamics within and patterns across human hosts at the level of ecological strains (12, 13), the macroecology of microbial communities across taxonomic and phylogenetic scales (14), and the dependency between correlations in abundance between community members and phylogenetic distance (15).

We identified and re-processed time-resolved human gut microbiome 16S rRNA datasets from healthy hosts with daily or near-daily sampling (16–19) (Supplemental Material; Table S1). We inferred the mean and CV of each ASV using sampling-aware maximum likelihood (Supplemental Material; Fig. S1). As typical abundances can vary by orders of magnitude in microbial communities, abundances were rescaled by their time-averaged mean and log-transformed, $y_i(t) \equiv \ln\left(\frac{x_i(t)}{\bar{x}_i}\right)$ (Fig. S2).

We consider the SLM as a lens to investigate empirical sojourn patterns, defined as follows:

$$\frac{dx_i}{dt} = \frac{x_i}{\tau_i}\left(1 - \frac{x_i}{K_i}\right) + \sqrt{\frac{\sigma_i}{\tau_i}}x_i \cdot \eta_i(t).$$ (1)

The SLM contains three parameters: (i) the timescale $\tau_i$ (inverse of the maximum growth rate), (ii) the carrying capacity $K_i$, and (iii) the strength of environmental noise $\sigma_i$. The term $\eta_i(t)$ introduces stochasticity into the equation as white noise: the expected value of is $\langle \eta_i(t) \rangle = 0$, and the time correlation is a delta function $\langle \eta_i(t)\eta_i(t') \rangle = \delta(t - t')$ (i.e., noise at time $t$ is uncorrelated with the noise at time $t'$).

We first investigated sojourn trajectories as consecutive observations where a community member deviated from and proceeded to return to its steady-state abundance (Fig. 1a). By calculating sojourn times for all ASVs in all datasets, we were able to obtain an empirical probability distribution of $\mathcal{T}$ (Fig. 1b). We first compared the empirical distribution to those obtained under two null models that both reflect the absence of temporal dynamics contributing to $\mathcal{T}$. Neither the null distribution obtained by (i) independently drawing observations from the gamma distribution (stationary distribution of the SLM) nor (ii) permuting time labels in the data reproduced the empirical distribution (Fig. 1b; Fig. S3). In contrast, the time-dependent solution of the SLM (function of $\tau$) captured the bulk of the empirical distribution, though we note the existence of a few high values of $\mathcal{T}$ that were not captured by the SLM. To assess whether $\mathcal{T}$ contains greater temporal information than previously investigated macroecological patterns, we examined the length of time that ASVs were consistently observed or unobserved, quantities known as the residence and return times that have been the subject of prior macroecological investigations (20–22). For each quantity, we calculated the divergence between the empirical and time-permuted null distributions, finding that $\mathcal{T}$ displayed greater divergence than the other two quantities in all hosts (Supplemental Material; Fig. S4 to S7). These results suggest that sojourn times have greater sensitivity to underlying temporal dynamics. We note that ASVs that are periodically absent (i.e., those used to calculate residence and return times) may reflect dynamics where abundances occasionally and rapidly become highly abundant (i.e., conditionally rare taxa) (23). For ASVs consistently observed throughout a given timeseries (i.e., those used to calculate sojourn times), we note that periods of atypically large abundances would be interpreted as sojourn trajectories.

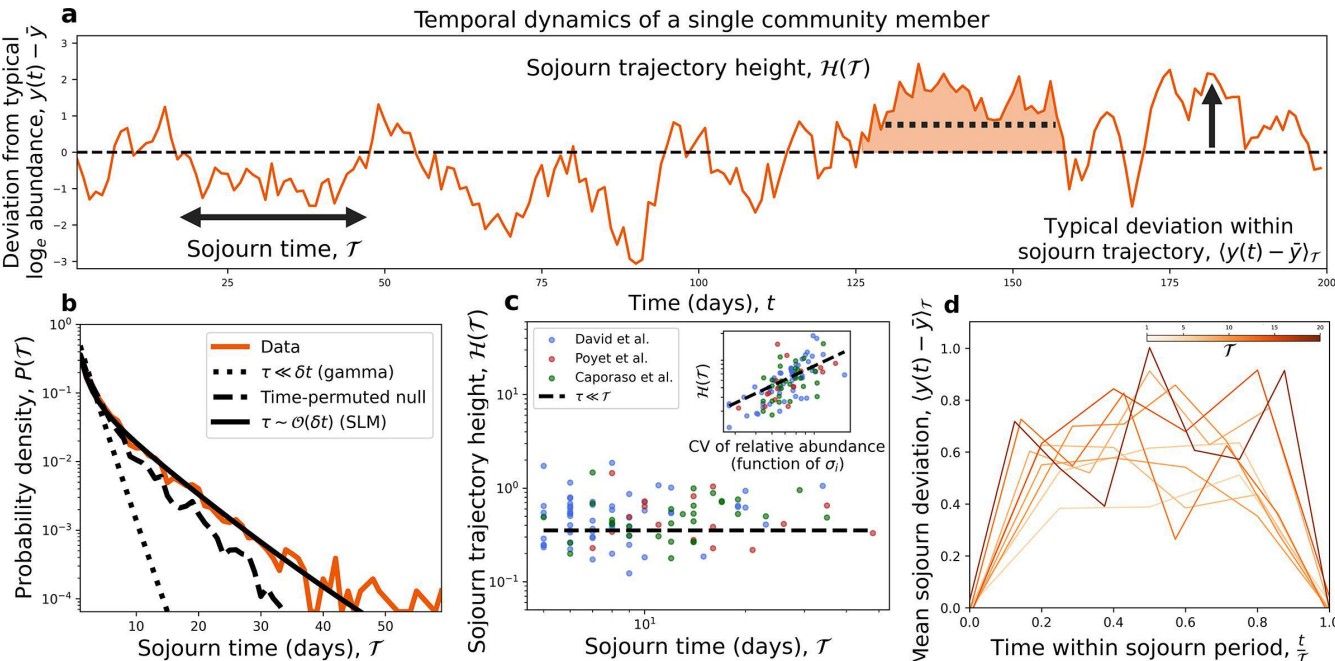

**FIG 1** Patterns of sojourn trajectories inform our understanding of microbial community dynamics. (a) We investigated the macroecology of sojourn trajectories using log-transformed relative abundances that have been rescaled by their mean ($y_i(t) \equiv ln\left(\frac{x_i(t)}{\bar{x}_i}\right)$). Consecutive fluctuations of $y_i(t)$ either above or below the time-averaged mean were investigated, focusing on three previously identified patterns (4, 5): (i) the distribution of sojourn times $p(\mathcal{T})$, (ii) the relationship between $\mathcal{T}$ and the height of a sojourn trajectory ($\mathcal{H}$), and (iii) how the mean deviation within a sojourn trajectory depends on $\mathcal{T}$. (b) The empirical $p(\mathcal{T})$) could not be captured by null distributions obtained by (i) assuming abundances are stationary (i.e., independent draws from a gamma distribution) nor (ii) permuting time labels. However, the distribution predicted by the SLM captured the bulk of the distribution ($\tau= 4$; equation S23). (c) We observed a lack of dependency between $\mathcal{T}$ and $\mathcal{H}$ across datasets, but a consistent relationship between the inverse strength of environmental noise and $\mathcal{H}$, results that are predicted by the SLM under the parameter limit where the growth rate is typically smaller than $\mathcal{T}$. Each dot represents a single sojourn trajectory from a given ASV. (d) Given that $\mathcal{H}$ is a constant, we predicted that the mean sojourn deviation would not vary with $\mathcal{T}$, which was observed in the data. The mean deviation was estimated by pooling over ASVs and sojourn trajectories with the same value $\mathcal{T}$. Predictions were obtained using ASV-specific values of $\sigma_i$.

We then investigated how $\mathcal{T}$ as a measure of length relates to the height under a sojourn trajectory [$\mathcal{H}(\mathcal{T})$; Fig. 1a], a dependency that has been previously characterized as a scaling relationship (4, 5) (Fig. 1a; Supplemental Material). This relationship determines whether and how different trajectories must be rescaled to have the same shape. In empirical data, we found that the height of a sojourn trajectory was effectively independent of its sojourn time [$\mathcal{H}(\mathcal{T}) \to \mathcal{H}$; Fig. 1c]. The absence of a relationship between the two quantities was predicted by the SLM when the timescale of growth is much smaller than the typical sojourn time ($\tau \ll \mathcal{T}$; Supplementary Material). Motivated by this result, we tested the prediction that under this parameter regime $\mathcal{H}$ would scale with the CV of an ASV (a function of $\sigma_i$), finding that the exponent predicted by the SLM sufficiently captured the empirical trend (inset of Fig. 1c).

The fact that $\mathcal{A}$ is independent of $\mathcal{T}$ informs our understanding of the typical deviation from the steady state within a sojourn trajectory, $\langle y_i(t) - \bar{y}_i \rangle_\mathcal{T}$. Prior theoretical efforts have found that a wide range of stochastic dynamics can be captured by a scaling relationship that depends on $\mathcal{T}^\alpha$ for a given exponent $\alpha$ (Supplemental Material) (4, 5). The absence of a relationship in Fig. 1c implies that $\alpha = 0$, meaning that sojourn deviations should exhibit similar trajectories for different values of $\mathcal{T}$ without additional rescaling. This prediction held, as empirical trajectories $\langle y_i(t) - \bar{y}_i \rangle$ did not systematically change with increasing $\mathcal{T}$ (Fig. 1d; Fig. S8).

Up to this point, we have investigated the similarities in the sojourn dynamics of microbial community members but have neglected how variation in fluctuations can shape typical sojourn times. Therefore, we examined the relationship between the mean

sojourn time (i.e., $\langle \mathcal{T} \rangle$) and both the mean abundance (function of $K_i$ and $\sigma_i$) and the coefficient of variation of abundance (solely a function of $\sigma_i$) (Fig. 2a). We focused on the dataset with the highest temporal resolution in order to estimate sojourn times $\mathcal{T}$ of the lowest possible values (Table S1) (17). The SLM predicted the absence of a relationship between $\langle \mathcal{T} \rangle$ and mean abundance, which held in the data (Supplemental Material; Fig. 2b; Fig. S9). The SLM also predicted the existence of a relationship between $\langle \mathcal{T} \rangle$ and the CV, which was also recapitulated in the data (Fig. 2c). These results indicate that the typical sojourn time in the gut is highly dependent on the CV, which is interpreted under the SLM as the strength of environmental noise that a community member experiences.

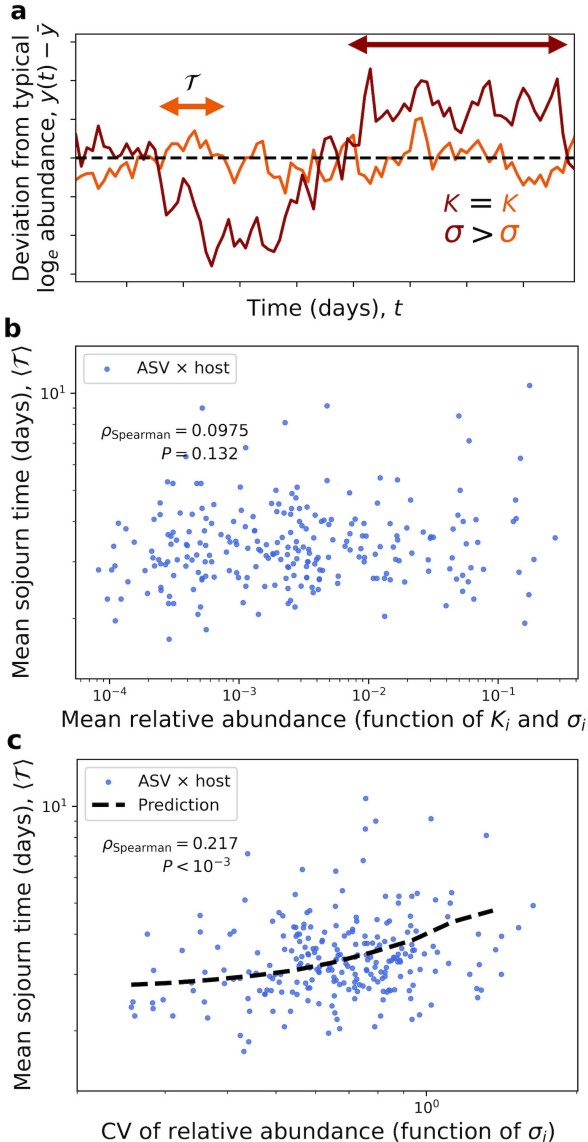

**FIG 2** Mean sojourn times are independent of mean abundance but dependent on the CV. (a) The SLM predicted that the mean sojourn time (i.e., $\langle \mathcal{T} \rangle$) of a community member would be independent of its mean relative abundance (i.e., $K_i$) but would increase with its CV (i.e., higher $\sigma_i$). (b) Consistent with our predictions, we found that mean relative abundance $\bar{x}_i$ (function of $K_i$ and $\sigma_i$) had no relationship with $\langle \mathcal{T} \rangle$. This absence can be interpreted as a lack of dependency of sojourn trajectories on the carrying capacity. (c) For the CV (solely a function of $\sigma_i$), a relationship existed which could be captured using predictions derived from the SLM. Spearman's rank correlation coefficient was used to assess the statistical significance of each relationship and predictions were obtained using ASV-specific values of $\sigma_i$.

Building on prior investigations into the temporal dynamics of microbial communities (9, 20, 21, 24, 25), this study focused on the macroecology of excursions from steady-state abundances by leveraging a theoretical framework from statistical physics (4, 5). We found that while the timescale of growth was necessary to recapitulate the distribution of sojourn times, sojourn trajectories were effectively scale-free with respect to sojourn time. Similarly, the mean sojourn time of an ASV could be predicted by the CV. These results can be interpreted under the SLM as excursions from the steady state reflecting the strength of environmental noise.

Due to the available data, this study focused on healthy hosts, though it would be of interest to extend our analyses to the scenario where microbiomes are effectively out of steady state, as such perturbations have the potential to impact stability (26). We note that our predictions were obtained from the SLM in a parameter limit where the model reduced to a form amenable to analytic solutions (Supplemental Material). This detail implies that while the SLM remains a viable minimal model for interpreting empirical microbial ecological dynamics, it may not be the sole model capable of explaining our observations. We emphasize that here the length of time between observations ($\delta t$) is similar to the intrinsic timescale of the system (i.e., the timescale of growth) due to the inherent limitations of fecal sample collection, in contrast to other systems where sojourn times have been investigated (6, 7). Alternative sojourn dynamics are likely to be found in community sampling scenarios where the timescale of growth exceeds that of sampling ($\tau \gg \delta t$), though models, such as the SLM may continue to provide useful interpretations.

## ACKNOWLEDGMENTS

We thank O. Mazzarisi for their feedback on the manuscript. Conceptual diagrams were created in BioRender (https://BioRender.com/0ec7ab7).

This work was supported by Fondo Italiano per la Scienza-FIS (CUP J53C23002290001; J.G.). Funders had no role in study design, data collection and analysis, decision to publish, or preparation of the manuscript.

W.R.S. and J.G. conceptualized the project, performed the derivations, and wrote the manuscript. W.R.S. performed all analyses.

## AUTHOR AFFILIATION

[1]Quantitative Life Sciences, The Abdus Salam International Centre for Theoretical Physics (ICTP), Trieste, Italy

## AUTHOR ORCIDs

William R. Shoemaker  http://orcid.org/0000-0003-0111-4838

## FUNDING

| Funder | Grant(s) | Author(s) |
| --- | --- | --- |
| Follie Italiane per la Scienza | CUP J53C2300229000 | Jacopo Grilli |

## AUTHOR CONTRIBUTIONS

William R. Shoemaker, Conceptualization, Data curation, Formal analysis, Investigation, Methodology, Project administration, Resources, Software, Validation, Visualization, Writing – original draft, Writing – review and editing | Jacopo Grilli, Funding acquisition, Project administration, Supervision, Writing – review and editing

## DATA AVAILABILITY

Public raw sequence data used in this study were obtained from previous studies (Table S1). Data processed for this study are available on Zenodo: https://doi.org/10.5281/zenodo.17809697. All codes are available on GitHub under a GNU General Public License: sojourn_macroeco.

## ADDITIONAL FILES

The following material is available online.

### Supplemental Material

**Supporting Information (mSystems01221-25-S0001.pdf).** Extended methods, supplemental figures, and supplemental table.

### Open Peer Review

**PEER REVIEW HISTORY (review-history.pdf).** An accounting of the reviewer comments and feedback.

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
