## [Reviewer comments · mSystems]

The macroecological dynamics of sojourn trajectories in the human gut microbiome

William Shoemaker and Jacopo Grilli

Corresponding Author(s): William Shoemaker, Abdus Salam International Centre for Theoretical Physics

Review Timeline:

Submission Date:	August 20, 2025
Editorial Decision:	October 28, 2025
Revision Received:	December 4, 2025
Accepted:	January 7, 2026

Editor: Luis Zaman

Reviewer(s): Disclosure of reviewer identity is with reference to reviewer comments included in decision letter(s). The following individuals involved in review of your submission have agreed to reveal their identity: María Rebolleda Gómez (Reviewer #1)

Transaction Report:

DOI: <https://doi.org/10.1128/msystems.01221-25>

Re: mSystems01221-25 (The macroecological dynamics of sojourn trajectories in the human gut microbiome)

Dear Dr. William Randolph Shoemaker:

Shoemaker and Grilli have presented an interesting treatment of statistical fluctuations in equilibrium densities of microbial communities (sojourn trajectories). I think their analysis will prove to be an important tool in the developing field of macroecological analysis of microbiomes.

The two reviewers have provided substantive feedback and have a few concerns I would also like to see addressed. Reviewer 1 notes that a more concrete connection between these dynamics and their biological interpretations would be useful (one example that came to my mind is how this relates to observations like conditionally rare taxa in microbiomes), and both reviewers would like to see some details about the SLM model in the main text. Reviewer 2 points out several claims of priority that I agree are not necessary.

Review 2 has some technical questions and suggestions that I believe will substantively improve the manuscript and its impact.

Revision Guidelines

Sincerely,
Luis Zaman
Editor

Reviewer #1 (Comments for the Author):

Summary

This manuscript formalizes a mathematical description of "sojourn trajectories" which describe temporary deviations from steady-state abundances. Often these trajectories are considered noise around a steady-state and therefore not worthy of further inspection, but they are a common occurrence across temporal series of microbiome data. Through their mathematical and statistical analysis, the authors were able to establish that: 1. the patterns observed are able to be explained by a stochastic logistic model of growth describing the changes in abundance of different species, 2. Sojourn area is independent of Sojourn time but 3. both Sojourn time and area depend on the strength of environmental noise. Both the analyses and results of this paper are important contributions to the field and provide an additional dimension of analyses of temporal data as well as clear expectations under this environmental noise driven regime.

Major comments

My main concern is that the biological context and implications of these results are not always clear in the text. I appreciate that some of the derivations and mathematical details are added in the supplementary material, but I think the paper could be strengthened and broaden if a brief explanation of the stochastic logistic model is moved to the main text. For example, I would move lines 93-111 from the supplement to the main text. Similarly, it would strengthen the paper to have a discussion briefly mentioning how can the results influence our ecological understanding of microbiomes (e.g. does the distribution of sojourn trajectories tells us something about environmental noise?). How does microbiome results compare to some of the other systems you introduced at the beginning, do you have any hypothesis of why that might be?

Is it possible to compare distributions between the hosts you have or is the sampling timing too variable across samples?

Minor comments and questions

Main text

Line 79 - Remove first sentence. Too repetitive.

Supplement

In converting the data to boolean vector - why is the cutoff at 1? What is the justification for that?

Line 75. Refs missing

Reviewer #2 (Comments for the Author):

In this short note, Shoemaker and Grilli examine the "sojourn trajectories" of microbial taxa - which describe the excursions above and below their estimated mean abundance - in timeseries data from the gut microbiome. They show that certain statistical features of these excursions can be captured by a simple stochastic logistic model (SLM) with three free parameters, representing their mean and coefficient of variation, as well as their autocorrelation time (τ). In particular, they find that (i) the autocorrelation time (τ) is significantly longer than the sampling interval (dt), and that (ii) the size of the excursion (dy) is independent of the sojourn time (T). Overall, I think this is a valuable contribution, which extends an existing body of work looking at macroecological properties of microbial timeseries. I have a few comments, questions, and suggestions for improvement:

(1) The abstract (line 17), introduction (line 38), and introduction (line 50) all contain a priority claim that (in my opinion) is somewhat overstated, relative to what has been done in the prior literature. Studies like David et al (Ref. 12), as well as the antibiotic perturbation literature (e.g. Ng et al Cell Host Microbe 2019, Xue et al (biorXiv 2023), and others) have investigated the excursions of microbial taxa after external perturbations, and studies like Ref. 15 have examined statistics (e.g. recovery time) that are conceptually similar to (though quantitatively different) the sojourn time here. Other studies (e.g. Chen et al PRX 2024) have examined temporal auto- and cross-correlation functions to infer the parameters of ecological models, which also implicitly uses information from sojourn trajectories.

I think the work would be strengthened by removing the priority claim from the abstract and importance sections (which read fine without it), and citing some of the above work in the intro to connect the present manuscript with the broader literature on this

topic.

(2) Given the central role that the SLM and its associated parameters (K , σ , and τ) play in the manuscript, it would be nice to include them somewhere for reference so that readers don't have to go back-and-forth to the SI. E.g. perhaps in a box in Fig. 1? Ideally, it would be nice to use a parameterization based on the mean and cv of the stationary distribution (rather than K and σ^2), since these are directly fit from the data. As far as I can tell, this leaves one free parameter (τ) that need to be inferred?

(3) A central finding of the manuscript is that the empirical distribution of sojourn times can reproduced by the stochastic logistic model but not a permuted model (i.e. that the autocorrelation time τ in the SLM is greater than the sampling time). However, I was a little confused about how this claim was assessed.

(3A) Where does the prediction line in Fig. 1B come from? I'm guessing that each taxon must be fit to the SLM to infer a corresponding τ . Where different τ s inferred for each taxon in each host? or was a single τ estimated for all taxa? (or all hosts?). In either case, what procedure was used to infer the best-fit value of τ ?

Similarly, I'm assuming that the species-specific value of σ was used to generate the theory curve as well, but it would be good to spell this out explicitly.

(3B) Which features of the empirical distribution should we be focusing on to assess how well the SLM is able to recapitulate the data? To my eyes, it looks as if there is an excess of really long excursions (and a deficit of really short excursions). Is the claim that these differences are not statistically significant (given the current sample size) so that we couldn't reject the SLM using a KS test or something similar? If so, please list the test. If the emphasis is instead on other qualitative features of the distribution (e.g. the median or the 90th percentile), it helpful to point out what the authors had in mind so that readers don't concentrate on the wrong thing.

(3C) Similarly, it would be nice to put the predictions of the null (permutation) model in Fig. 1B as well so that we can directly compare the two models against the data. (Note: the permutation model seems preferable to the geometric model here, since if I understand correctly, the former accounts for heterogeneity in sampling intervals while the latter does not?)

(4) One of the other claims in the abstract is that the typical sojourn time is independent of the mean abundance. Given its prominent role, it would be good to include the data for this finding in the main text, rather than a supplemental Fig.

However, I was a little confused how this prediction is related to the predictions of the SBM. If I understand correctly, the SBM predicts that the sojourn trajectory should be independent of the mean only after conditioning on the value of σ (and τ). But it's possible that σ could scale with the mean in a systematic way. In fact, some previous studies like Ji et al have shown that the relationship between the mean and variance follow a form of Taylor's law with a power < 2 . I am confused how this finding could be consistent with the claim above, given that the authors have shown that the sojourn distribution *does* depend on the coefficient of variation. It would be nice if the authors could comment on how their results relate to previous findings on Taylor's law in these data.

(5) I was confused about the results on the area statistic, $A(T)$, until later finding out in the SI that $A(T)$ is actually defined as the average *size* of the deviation (i.e. the area in Fig. 1 divided by T). Everything made more sense after that, since an OU process has a typical "height" of an excursion, independent of its length.

I would strongly advocate for either (i) reporting the actual area (as illustrated in Fig. 1) or (ii) calling $A(T)$ the average size instead, perhaps switching the notation to something like $S(T)$ or $\bar{y}(T)$?

(6) Fig. 1D looks like it has many fewer sojourn trajectories than Fig. 1C... why is this the case? (from the SI it sounded like the restrictions on the number of timepoints was greater for panel C than panel D). More broadly, I was a little confused about the claims on lines 95-98. My understanding of the theoretical result is that the typical *height* of the deviation is independent of T , but the shape itself retains some residual dependence on T , since there is always some initial phase (of size τ) for the trajectory to go from 0 to its maximum typical value?

Point-by-point response

Reviewer #1

Summary

This manuscript formalizes a mathematical description of "sojourn trajectories" which describe temporary deviations from steady-state abundances. Often these trajectories are considered noise around a steady-state and therefore not worthy of further inspection, but they are a common occurrence across temporal series of microbiome data. Through their mathematical and statistical analysis, the authors were able to establish that: 1. the patterns observed are able to be explained by a stochastic logistic model of growth describing the changes in abundance of different species, 2. Sojourn area is independent of Sojourn time but 3. both Sojourn time and area depend on the strength of environmental noise. Both the analyses and results of this paper are important contributions to the field and provide an additional dimension of analyses of temporal data as well as clear expectations under this environmental noise driven regime.

Major comments

My main concern is that the biological context and implications of these results are not always clear in the text. I appreciate that some of the derivations and mathematical details are added in the supplementary material, but I think the paper could be strengthened and broadened if a brief explanation of the stochastic logistic model is moved to the main text. For example, I would move lines 93-111 from the supplement to the main text.

- **Response:** We thank the reviewer for their suggestions. In the revised manuscript we have moved lines 93 - 111 from the supplement to the main text (now lines 65-70).

Similarly, it would strengthen the paper to have a discussion briefly mentioning how these results influence our ecological understanding of microbiomes (e.g. does the distribution of sojourn trajectories tell us something about environmental noise?). How do microbiome results compare to some of the other systems you introduced at the beginning, do you have any hypothesis of why that might be?

- Regarding how our results influence our understanding of the ecology of the microbiome, we elected to compare our sojourn time estimates to two other timescales: the length of time where an ASV is consistently observed (i.e., residence time) and unobserved (i.e., return time). Distributions of these timescales have been previously examined in the microbial ecology literature (Ji et al., *Nat. Microbiol.*, 2020; Ho et al., *eLife*, 2022), where it has been noted that their distributions do not noticeably differ from null distributions obtained from the same time-permuted procedure used in this manuscript (Wang and Liu, *bioRxiv*, 2021; Wang and Liu, *Phys. Rev. Res.*, 2023). To determine whether sojourn time reflects temporal dynamics to a greater extent than residence and return times we calculated the divergence between the observed and null

distributions for each of the three measures of time. We found that sojourn time distributions were consistently more diverged than residence and return distributions in all hosts (Figs. S4-S7), meaning that in this manuscript we have identified a measurable timescale that displays greater sensitivity to temporal dynamics.

We have added a brief discussion to the manuscript on how, under the SLM, sojourn trajectories reflect environmental noise. Namely, deviations from steady-state values indicate stronger environmental noise. Regarding comparisons between different types of environments, if the SLM continued to serve as a valid model then differences in sojourn trajectories of a specified community member could be interpreted as environment-specific noise strengths. We emphasize how the temporal dynamics one can observe are an outcome of the intrinsic timescale of the community (i.e., τ) and the sampling resolution.

We summarize the above points in the revised manuscript (Main lines 81-89, 123-126, 132-134, 142-148; Supplemental Material 129-150).

Is it possible to compare distributions between the hosts you have or is the sampling timing too variable across samples?

- **Response:** We thank the reviewer for raising an interesting question regarding the variation in sojourn times *across hosts*. The mean number of days between sampling events is comparable across hosts (Table S1), so it is in principle possible. However, not all hosts contain the same ASVs, making comparison difficult. We also note that we expect the dynamics of a given ASV in different hosts to be *independent*, meaning that we would effectively be comparing two sojourn time distributions with no between-host correlation (Supplemental Material 74-77).

Minor comments and questions

Main text

Line 79 - Remove first sentence. Too repetitive.

- **Response:** We have removed the sentence.

Supplement

In converting the data to a boolean vector - why is the cutoff at 1? What is the justification for that?

- **Response:** We have corrected this typo in the revised manuscript to specify that we are classifying observations based on whether they were greater or less than the expected value (Supplemental Material 64-69).

Line 75. Refs missing

- **Response:** The references have been added.

Reviewer #2:

In this short note, Shoemaker and Grilli examine the "sojourn trajectories" of microbial taxa - which describe the excursions above and below their estimated mean abundance - in timeseries data from the gut microbiome. They show that certain statistical features of these excursions can be captured by a simple stochastic logistic model (SLM) with three free parameters, representing their mean and coefficient of variation, as well as their autocorrelation time (τ). In particular, they find that (i) the autocorrelation time (τ) is significantly longer than the sampling interval (dt), and that (ii) the size of the excursion (dy) is independent of the sojourn time (T). Overall, I think this is a valuable contribution, which extends an existing body of work looking at macroecological properties of microbial timeseries. I have a few comments, questions, and suggestions for improvement:

(1) The abstract (line 17), introduction (line 38), and introduction (line 50) all contain a priority claim that (in my opinion) is somewhat overstated, relative to what has been done in the prior literature. Studies like David et al (Ref. 12), as well as the antibiotic perturbation literature (e.g. Ng et al Cell Host Microbe 2019, Xue et al (biorXiv 2023), and others) have investigated the excursions of microbial taxa after external perturbations, and studies like Ref. 15 have examined statistics (e.g. recovery time) that are conceptually similar to (though quantitatively different) the sojourn time here. Other studies (e.g. Chen et al PRX 2024) have examined temporal auto- and cross-correlation functions to infer the parameters of ecological models, which also implicitly uses information from sojourn trajectories.

I think the work would be strengthened by removing the priority claim from the abstract and importance sections (which read fine without it), and citing some of the above work in the intro to connect the present manuscript with the broader literature on this topic.

- **Response:** We thank the reviewer for suggesting relevant literature. We have removed the priority claim from the main text and now only briefly state in the Supplement that these three specific patterns of sojourn trajectories have not, to our knowledge, been previously examined in ecological systems.

We now discuss and cite Ng et al. and Xue et al. at the start of the manuscript, focusing on how experimentally-imposed perturbations differ from fluctuations.

By “recovery time” we assume the reviewer is referring to the “return time” distributions examined by Ji et al. (Nat. Microbiol., 2020). Review One also asked how sojourn times compared to the timescales studied by Ji et al. In our revised manuscript we calculate time-permuted null distributions of 1) sojourn times, 2) return times, and 3) residence times, the latter two being objects of study in Ji et al. (Nat. Microbiol., 2020). We find that empirical sojourn time distributions deviate from their null distribution to a greater extent than return and residence times, indicating that they are a stronger reflection of temporal dynamics (Main 81-89; Supplemental Material 129-150; Figs. S4-S7).

Regarding temporal auto/crosscorrelation, we briefly discuss how these measures relate to sojourn trajectories in the context of the SLM (Supplemental Material lines 179-189).

(2) Given the central role that the SLM and its associated parameters (K , σ , and τ) play in the manuscript, it would be nice to include them somewhere for reference so that readers don't have to go back-and-forth to the SI. E.g. perhaps in a box in Fig. 1?

- **Response:** Following the suggestion from Reviewer 1, we have moved the overview of the SLM from the SI to the main manuscript (Main lines 65-70).

Ideally, it would be nice to use a parameterization based on the mean and cv of the stationary distribution (rather than K and σ^2), since these are directly fit from the data. As far as I can tell, this leaves one free parameter (τ) that need to be inferred?

- **Response:** In the revised manuscript we now consistently use parameterization based on the mean and CV throughout the text and in figure legends (e.g., legends of Figs. 1d, 2).

(3) A central finding of the manuscript is that the empirical distribution of sojourn times reproduced by the stochastic logistic model but not a permuted model (i.e. that the autocorrelation time τ in the SLM is greater than the sampling time). However, I was a little confused about how this claim was assessed.

Response: To evaluate whether the empirical sojourn time distribution reflected temporal dynamics we compared it to two null distributions. We first derived a prediction from a null model where the time between samples is much greater than autocorrelation time τ . In this regime a pair of samples generated by the SLM will effectively be independent gamma-distributed random variables (Supplemental Material lines 78-92; Eq. S8). As an alternative method, we also obtained a null distribution by permuting the time labels in the data, conserving the empirical distribution of abundances while destroying any information about the temporal dynamics

We clarify our above strategy in the revised main manuscript (lines 74-79) and the supplement (lines 69-73; 78-92) and have added both null models to Fig. 1b.

(3A) Where does the prediction line in Fig. 1B come from? I'm guessing that each taxon must be fit to the SLM to infer a corresponding tau. Were different taus inferred for each taxon in each host? or was a single tau estimated for all taxa? (or all hosts?). In either case, what procedure was used to infer the best-fit value of tau?

Response: Inferring the parameter tau from empirical relative abundances using the SLM is a non-trivial task. At present there is no closed-form maximum likelihood estimator for the SLM. This is partially due to there being no close-form time-dependent solution of the SLM (see S5 Text in Shoemaker et al., *Plos. Comp. Bio.*, 2025). In contrast, the strength of environmental noise and the carrying capacity could be inferred for each ASV, as they are present in the gamma distribution as the stationary limit of the SLM (Eq. S12, S13) where a sampling-aware maximum likelihood procedure could be numerically implemented (Supplemental Material lines 33-46, Eq. S2).

We are actively working on the per-ASV inference of tau using the SLM. However, because ASV-specific estimates of tau do not factor into our demonstration of the validity of the sojourn trajectory scaling relationship (i.e., Fig. 1c,d; patterns 1 and 2 in the Supplement), we elected to choose a single value of tau for all ASVs. We found that a value of tau = 4 days captured the bulk of the empirical sojourn time distribution (Supplemental Material line 229 - 231; Fig. 1b).

Similarly, I'm assuming that the species-specific value of sigma was used to generate the theory curve as well, but it would be good to spell this out explicitly.

Response: We have clarified the use of ASV-specific values of the strength of environmental noise in both the supplement (Supplemental Material lines 202-204) as well as the captions of Figs. 1 and 2.

(3B) Which features of the empirical distribution should we be focusing on to assess how well the SLM is able to recapitulate the data? To my eyes, it looks as if there is an excess of really long excursions (and a deficit of really short excursions). Is the claim that these differences are not statistically significant (given the current sample size) so that we couldn't reject the SLM using a KS test or something similar? If so, please list the test. If the emphasis is instead on other qualitative features of the distribution (e.g. the median or the 90th percentile), it helpful to point out what the authors had in mind so that readers don't concentrate on the wrong thing.

Response: We thank the reviewer for pointing out the discrepancy between the distribution predicted by the SLM and the empirical distribution in our original

submission. This observation caused us to revisit our code, where we noticed that we neglected to account for sojourn time of the data being discrete (i.e., # days as an integer) whereas the predicted distribution modeled sojourn time as a continuous random variable (Eq. S23). It is necessary for the variable to have the same data type for both distributions to ensure a proper comparison. We derive a form of the predicted distribution where sojourn time is a discrete random variable and reference the equation in the legend of Fig. 1.

Correcting this original oversight resulted in the predicted and empirical distributions bearing greater resemblance, reducing the gap between the two distributions for long sojourn times and strengthening our claim that the SLM captures the bulk of the empirical distribution. A few rare, long excursions do remain, which we point out in the revised manuscript (Main lines 79-81).

(3C) Similarly, it would be nice to put the predictions of the null (permutation) model in Fig. 1B as well so that we can directly compare the two models against the data. (Note: the permutation model seems preferable to the geometric model here, since if I understand correctly, the former accounts for heterogeneity in sampling intervals while the latter does not?)

Response: We thank the reviewer for their suggestion. Both the permutation and analytic null models are now included in Fig. 1b. We now state the reviewer's observation about sampling intervals in the supplement (lines 72-73). In the revision we sought to limit the influence of heterogeneous sampling intervals by selecting subsets of timeseries with near-daily sampling (Supplemental Material lines 22-31)

(4) One of the other claims in the abstract is that the typical sojourn time is independent of the mean abundance. Given its prominent role, it would be good to include the data for this finding in the main text, rather than a supplemental Fig.

Response: In the revised manuscript we now include the relationship between the mean abundance and mean sojourn time in Fig. 2.

However, I was a little confused how this prediction is related to the predictions of the SLM. If I understand correctly, the SLM predicts that the sojourn trajectory should be independent of the mean only after conditioning on the value of sigma (and tau).

Response: The mean sojourn time is independent of the mean abundance. No additional conditioning on sigma or tau is required. Under the SLM the mean abundance is a function of both the strength of environmental noise (sigma) and the carrying capacity (K). Because the mean sojourn time (Eq. S26) does not contain K, we cannot define the mean sojourn time as a function of the relative abundance. In contrast, the CV

of abundance is solely a function of sigma, allowing us to define the mean sojourn time as a function of the CV of abundance. We now state these points in the revised manuscript (Supplemental Material lines 243 - 247).

But it's possible that sigma could scale with the mean in a systematic way. In fact, some previous studies like Ji et al have shown that the relationship between the mean and variance follow a form of \approx law with a power < 2 . I am confused how this finding could be consistent with the claim above, given that the authors have shown that the sojourn distribution **does** depend on the coefficient of variation. It would be nice if the authors could comment on how their results relate to previous findings on Taylor's law in these data.

Response: The independence of the mean sojourn time from the mean abundance holds on a **per-ASV** basis, whereas Taylor's Law describes the dependency of the variance on the mean **across ASVs**. This distinction means that the absence of a relationship between mean sojourn time and mean abundance will hold regardless of the existence of Taylor's Law.

The question of whether sigma scales with mean abundance can be rephrased as asking whether the CV of abundance scales with mean abundance. We examined this relationship for each host and found an absence of a relationship in all but one host (Fig S9). We clarify the above points in the revised manuscript (Supplemental Material lines 247-251).

(5) I was confused about the results on the area statistic, $A(T)$, until later finding out in the SI that $A(T)$ is actually defined as the average **size** of the deviation (i.e. the area in Fig. 1 divided by T). Everything made more sense after that, since an OU process has a typical "height" of an excursion, independent of its length.

I would strongly advocate for either (i) reporting the actual area (as illustrated in Fig. 1) or (ii) calling $A(T)$ the average size instead, perhaps switching the notation to something like $S(T)$ or $y_{\text{bar}}(T)$?

Response: We thank the reviewer for their feedback. We note that the quantity $A(T)$ represents the *area* under the curve, not the average. The integral over a single sojourn trajectory of length T for a given ASV is defined as Eq. S10 and visualized in Fig. 1a. This integral is not divided by T , meaning that the formula represents the area. We estimate the area in the empirical data via numerical integration and do not divide the resulting quantity by T . We also derive $A(T)$ for the SLM without dividing by T (Eq. S37). We clarify the meaning of the above integral in the revised manuscript (Main lines 95, 99; Supplemental Material lines 105-107).

(6) Fig. 1D looks like it has many fewer sojourn trajectories than Fig. 1C... why is this the case? (from the SI it sounded like the restrictions on the number of timepoints was greater for panel C than panel D).

Response: Each dot of the same color in Fig. 1C represents a single sojourn trajectory from a given ASV. We now clarify this point in the caption of Fig. 1.

In Fig. 1d the y-axis is defined as the expected value of the deviation from the mean for a sojourn trajectory of length T based on the previously established sojourn scaling relationship (Baldassarri et al., 2003; Colaiori et al., 2004). We calculated this value by taking the mean over different sojourn trajectories, ASVs, and hosts constrained on them all having the same sojourn time T (Supplemental Material lines 122-128; caption of Fig. 1). We understand that readers may want to inspect individual deviations rather than their mean, which we provided in the supplement by plotting individual trajectories for separate hosts (Fig. S8).

More broadly, I was a little confused about the claims on lines 95-98. My understanding of the theoretical result is that the typical *height* of the deviation is independent of T , but the shape itself retains some residual dependence on T , since there is always some initial phase (of size τ) for the trajectory to go from 0 to its maximum typical value?

Response: Lines 95-98 of the original manuscript are referring to the relationship between sojourn time (T) and the *area* of the curve under a sojourn trajectory ($A(T)$). We did not examine the *height* of the sojourn trajectory. We found that the area of the sojourn trajectory was independent of T and appeared to follow the form predicted in the limit where the typical sojourn time is greater than the timescale of growth (inset of Fig. 1c; Eq. S37). We note that neither of the two parameter regimes for area nor the scaling relationship for the expected sojourn trajectory (Eq. S38, S39) explicitly depend on τ . In addition, the independence of $A(T)$ from T means that *the sojourn trajectory does not depend on T* . This is because the absence of this relationship implies that the exponent eq. S38 is equal to zero (i.e., $\alpha = 0$), resulting in T only contributing to the reduced equation in a dimensionless form (i.e., $f(t/T)$ does not explicitly depend on T). We have rewritten the manuscript and SI to reflect the above response (main manuscript 107-113; Supplement lines 280-296).

While there is noise in the data due to there being a finite number of reads, we believe it is fair to claim that the shape of the deviation trajectory defined as the expected deviation $\langle y(s^*T) - \bar{y} \rangle_T$ is independent of T . Values of the expected deviation do not appear to be sorted by their value of T , in the sense that lower values of T tended to be above higher values of T or vice versa (Main lines 110-113).

References

- Baldassarri, A., Colaiori, F., & Castellano, C. (2003). Average shape of a fluctuation: Universality in excursions of stochastic processes. *Physical review letters*, 90(6), 060601.
- Colaiori, F., Baldassarri, A., & Castellano, C. (2004). Average trajectory of returning walks. *Physical Review E—Statistical, Nonlinear, and Soft Matter Physics*, 69(4), 041105.
- Ho, P. Y., Good, B. H., & Huang, K. C. (2022). Competition for fluctuating resources reproduces statistics of species abundance over time across wide-ranging microbiotas. *Elife*, 11, e75168.
- Ji, B. W., Sheth, R. U., Dixit, P. D., Tchourine, K., & Vitkup, D. (2020). Macroecological dynamics of gut microbiota. *Nature microbiology*, 5(5), 768-775.
- Ng, K. M., Aranda-Díaz, A., Tropini, C., Frankel, M. R., Van Treuren, W., O'Loughlin, C. T., ... & Huang, K. C. (2019). Recovery of the gut microbiota after antibiotics depends on host diet, community context, and environmental reservoirs. *Cell host & microbe*, 26(5), 650-665.
- Shoemaker, W. R., Sánchez, Á., & Grilli, J. (2025). Macroecological patterns in experimental microbial communities. *PLOS Computational Biology*, 21(5), e1013044.
- Wang, X. W., & Liu, Y. Y. (2021). Characterizing scaling laws in gut microbial dynamics from time series data: caution is warranted. *bioRxiv*, 2021-01.
- Wang, X. W., & Liu, Y. Y. (2023). Origins of scaling laws in microbial dynamics. *Physical Review Research*, 5(1), 013004.
- Xue, K. S., Walton, S. J., Goldman, D. A., Morrison, M. L., Verster, A. J., Parrott, A. B., ... & Relman, D. A. (2023). Prolonged delays in human microbiota transmission after a controlled antibiotic perturbation. *bioRxiv*.

Re: mSystems01221-25R1 (The macroecological dynamics of sojourn trajectories in the human gut microbiome)

Dear Dr. William Randolph Shoemaker:

The authors have provided a very thoughtful and detailed response to the original reviewers' comments. I am excited to see this short (but rigorous) Observation be accepted for publication!

Your manuscript has been accepted, and I am forwarding it to the ASM production staff for publication. Your paper will first be checked to make sure all elements meet the technical requirements. ASM staff will contact you if anything needs to be revised before copyediting and production can begin. Otherwise, you will be notified when your proofs are ready to be viewed.

Sincerely,
Luis Zaman
Editor
mSystems

Reviewer #2 (Comments for the Author):

I'd like to thank the authors for their work in revising the manuscript. Most of my comments are addressed. The one remaining issue is the definition of the area $A(T)$ (Comment X) from last round of review. I don't mean to dwell on this too much, since it's more a language issue than anything else, but I still worry that it could lead to confusion for other readers like myself. From the illustration in Fig. 1A, I think most readers would expect the area $A(T)$ to be defined as

$$A(T) = \int_0^T (x(t) - x(0)) dt.$$

Eq. S9 in the SI defines it slightly differently, using the change of variables $s = t/T$, so that

$$A_{S9}(T) = \int_0^1 (x(sT) - x(0)) ds = 1/T * A(T).$$

This is what I meant last time when I said that Eq. S9 is the average *size* of the deviation (i.e. the area divided by T), due to the

choice of integration variable. (Incidentally, this appears to be absent in Eq. S9, but I'm assuming it's ds like Eq. S10.)

This distinction is important for interpreting the discussion on line 110. Using the heuristic that area = height x width, a longer width (T) would necessarily lead to a larger area $A(T)$, unless the height is smaller to compensate. As far as I can understand, that's not really what's going on here - instead, the height saturates (independent of T) so that the true area (as illustrated in Fig. 1A) would scale linearly with time.

I'll leave it to the authors to decide how to handle this, but I do think it would be less confusing for readers if the area was defined without the "s" integration variable to start with, since it more closely matches Fig. 1 (and the intuitive notion of the word area).